

**In situ $^{10}$Be modeling and terrain analysis constrain subglacial**
**quarrying and abrasion at Jakobshavn Isbræ, Greenland**
Brandon L. Graham[1], Jason P. Briner[1], Nicolás E. Young[2], Allie Balter-Kennedy[2], Michele
Koppes[3], Joerg M. Schaefer[2], Kristin Poinar[1] and Elizabeth K. Thomas[1]
[1]Department of Geology, University at Buffalo, Buffalo NY USA
[2]Lamont-Doherty Earth Observatory, Columbia University, USA
[3]University of British Columbia, Vancouver, BC, Canada
*Correspondence to*: Jason P. Briner (jbriner@buffalo.edu)



**Abstract.** Glacial erosion creates diagnostic landscapes and vast amounts of sediment. Yet, knowledge about the
rate by which glaciers erode and sculpt bedrock and the proportion of quarried (plucked) versus abraded material is
limited. To address this, we quantify subglacial erosion rates and constrain the ratio of quarrying to abrasion during
the 19th/20th century overriding of a bedrock surface fronting Jakobshavn Isbræ, Greenland, by combining [10]Be
analyses, a digital terrain model, and field observations. Cosmogenic [10]Be measurements along a 1.2-m-tall quarried
bedrock step reveal a triangular wedge of quarried rock. Using individual [10]Be measurements from abraded surfaces
across the study area, we derive an average abrasion rate of $0.13\pm0.08$ mm yr$^{-1}$. By applying this analysis across a
~1.33 km$^2$ study area, we estimate that the Greenland Ice Sheet quarried $378\pm45$ m$^3$ and abraded $322\pm204$ m$^3$ of
material at this site. These values result in an average total erosion rate of $0.26\pm0.16$ mm yr$^{-1}$ with abrasion and
quarrying contributing in roughly equal proportions within uncertainty. Additional cosmogenic [10]Be analysis and
surface texture mapping indicate that many lee steps are relict from the prior glaciation and were not re-quarried
during the recent overriding event. These new observations of glacier erosion in a recently exposed landscape
provide one of the first direct measurements of quarrying rates and indicate that quarrying accounts for roughly half
of total glacial erosion in representative continental shield lithologies.

**1. Introduction**

Distinctive features of glacier erosion characterize most glaciated regions, ranging from polished bedrock

surfaces to overdeepened fjords. Additionally, vast amounts of sediment are produced via glacial erosion. The
Greenland Ice Sheet accounts for a disproportionate delivery of sediment to the oceans, which impacts marine
ecosystems and carbon sequestration (e.g., Overeem et al., 2017). The two dominant mechanisms of glacier erosion
are subglacial quarrying and abrasion (Alley et al., 2019). Quarrying occurs when bedrock blocks are episodically
entrained and removed by overriding glaciers (e.g., Hallet, 1996; Iverson, 2012, Koppes, 2022). Abrasion occurs via
the gradual wearing down of bedrock surfaces as rock fragments are entrained and pressed into the bed by sliding
ice (Hallet, 1979; Iverson, 1990, Koppes, 2022). The rate at which each of these processes occur is dictated by rock
properties (e.g., Matthes, 1930; Dühnforth et al., 2010; Krabbendam and Glasser, 2011), glacio-hydraulic factors
(e.g., Egholm et al., 2012; Zoet et al., 2013; Anderson, 2014) and climate (e.g., Cook et al., 2020; Koppes 2022).
Although the result of the work done by glaciers on landscapes is dramatic, observational datasets that constrain
how quickly landscapes are modified by ice remain sparse (Alley et al., 2019).

Despite considerable challenges in observing erosional processes occurring under ice, our understanding of

subglacial erosion rates continues to expand. Total glacial erosion rates (i.e., abrasion + quarrying) have been
inferred using a variety of approaches (e.g., Hallet et al., 1996; Herman et al., 2021; Koppes, 2022) and are found to
generally fall between 0.01 and ≥1 mm yr$^{-1}$; however, higher rates have been measured on short (annual to decadal)
timescales (e.g., Koppes and Montgomery, 2009; Cowton et al., 2012). Attempts at separating the components of
quarrying and abrasion have been made based on sediment flux measurements (e.g., Loso et al., 2004; Riihimaki et
al., 2005), cosmogenic-nuclide inversions across subglacial bedforms (e.g., Briner and Swanson, 1998) and



theoretical considerations related to sparsely vs. intensely fractured bedrock (e.g., Anderson, 2014). To date,
measurements that isolate the eroded rock volume that can be attributed to quarrying are rare.

Here, we quantify subglacial erosion at a site that experienced a well constrained advance-retreat cycle of

Jakobshavn Isbræ, a major outlet glacier in West Greenland (Fig. 1). We partition total erosion into abrasion and
quarrying by pairing cosmogenic $^{10}$Be measurements with analysis of a high-resolution terrain model and field
mapping of bedrock surface textures. We model the accumulation of cosmogenic $^{10}$Be that we measured across a
quarried bedrock step to reconstruct the surface profile of the removed material, and the abrasion depth in adjacent
surfaces. Our data allow us to identify which bedrock steps experienced quarrying during the most recent advance of
the ice versus those unaltered since the prior glaciation. We thus calculate the volume of rock removed during the
recent overriding event by abrasion and by quarrying, and estimate the average erosion rate of each over the duration
of glacier overriding.

**2. Study area**

The north and south branches of Jakobshavn Isbræ merged and extended westward ~35 km within

Jakobshavn Isfjord to attain the "historical limit," which was observed in the fjord in 1850 CE (Fig. 1; Weidick and
Bennike, 2007). Adjacent to Jakobshavn Isfjord, the historical limit is represented by a prominent moraine
demarcating the extent of the "historical advance," which more or less coincides with the Little Ice Age. Prior
authors calculated (1) the timing of deglaciation following the last glacial maximum to the historical limit at 7500 yr
ago (Young et al., 2011; Balter-Kennedy et al., 201) and to the present ice position by 7400 yr ago, and (2) the
advance (~1790 CE) and retreat (2010 AD; a duration of 220±5 yr) of Jakobshavn Isbræ at the study site (Briner et
al., 2011; Young et al. 2016; Balter-Kennedy et al., 2021; Fig. 1). It is thought that Jakobshavn Isbræ receded during
the Holocene deglaciation to a position ~20 km inland of the present ice margin (Weidick et al., 1990; Kajanto et al.,
2020). Our study builds on Young et al. (2016) and Balter-Kennedy et al. (2021), who utilized cosmogenic $^{10}$Be
measurements to quantify total subglacial erosion rates of the gneissic bedrock in this area (Fig. 1).

**3. Methods**

In August 2018, we investigated a bedrock forefield adjacent to the north branch of Jakobshavn Isbræ that

emerged from beneath the ice between 2008 and 2010 based on available satellite imagery (Balter-Kennedy et al.,
2021). The surface of glacially abraded and quarried bedrock exhibits pristine features of glacial erosion (Fig. 2).
We measured ice-flow orientations, noted rock surface texture (variations in texture are accompanied by tonal
differences in the color of rock surfaces), used drone imagery to generate a high-resolution digital terrain model, and
collected samples for cosmogenic $^{10}$Be measurements.

Two stoss and lee landforms were chosen for detailed cosmogenic $^{10}$Be analysis, with the goal of

characterizing quarrying volume and timing. We chose one landform (Location A; Fig. 3A) to (1) estimate the
dimensions of the bedrock removed based on the geometry of a quarried divot, where there is a sharp transition from
rough to abraded surface texture surrounding the quarried zone, and (2) use $^{10}$Be concentrations in samples collected
from the quarried divot to reconstruct the profile of the pre-quarried surface. We created a 3D forward model of



cosmogenic [10]Be production to estimate the shape of the quarried material (single or multiple blocks) at location A.
The fundamental set-up of our conceptual model is shown in Figure 4. At another site (Location B; Fig. 3B), where
two adjacent lee steps have different surface textures, we measured the [10]Be concentration at the base of each step to
test our hypothesis that the variations in surface texture relate to quarrying during the historical overriding versus the
prior glaciation.

**3.1 Field sampling for [10]Be analysis**
At Location A, we nearly continuously sampled every ~5 cm from the top of the vertical face (0 cm, at the edge of
the stoss surface) to 69 cm depth, then we skipped down to sample the base of the lee face at 115 cm. We measured
[10]Be concentrations in five samples on the lee face: the top of the lee cliff ("surface"), from 12-15 cm, 30-33 cm, 65-
69 cm and 110-115 cm at the base. Wide, thin samples were collected (30 cm W x 3-5 cm H x 2-4 cm D) to
optimize the quartz mass within a narrow depth range and to minimize depth integration. We also collected three
samples along the horizontal floor, two from within the quarried scar and one beyond the distal edge of the quarried
scar from a polished surface (Fig. 3). At Location B, we collected one sample from the base of the lee cliff from
each zone (Fig. 3). All samples were collected with a combination of Hilti brand angle grinder with 5-inch diameter
diamond bit blades, and hammer and chisel. At all sampling locations, field observations of topographic shielding
were collected using a Brunton compass. Location and elevation were collected with a GPS time averaging smart
phone application with ±5 m accuracy.

**3.2 Terrain analysis and surface textures**
Aerial imagery was collected with a DJI Mavic Pro unmanned aerial vehicle (UAV) with continuous and
overlapping nadir imagery acquired using DJI smartphone app software. Maps Made Easy
(www.mapsmadeeasy.com; last access: April 26, 2023) was used to generate orthoimagery and a digital elevation
model (DEM) of the field area using structure from motion principles (Graham, 2023). Mosaic imagery was used as
a base layer for field mapping three surface roughness categories of the stoss and lee landforms based on the degree
of freshness (1: freshly exposed surfaces with minor grain-to-grain relief and no apparent abrasion, 2: lightly
abraded, 3: heavily abraded and polished). We also observed that the fresh-appearing surfaces exhibited darker
surface colors, and that smoother surface textures exhibited lighter surface colors. The orientation of ice flow
indicators consisting of striae, gouges, chatter marks, and crescentic gouges were measured using a compass.

**3.3 Beryllium-10 Laboratory Methods**
All physical rock processing and isolation of quartz for [10]Be analysis was performed at the University at Buffalo
Cosmogenic Isotope Laboratory (Corbett et al., 2016; Kohl & Nishiizumi, 1992). Pure quartz was processed at the
Lamont-Doherty Earth Observatory cosmogenic dating laboratory following established beryllium extraction
procedures. We processed eight samples from Location A, and two samples from Location B. AMS measurements
of [10]Be/[9]Be were performed at the Center for Accelerator Mass Spectrometry at Lawrence Livermore National
Laboratory (LLNL-CAMS), with references relative to the 07KNSTD standard of known [10]Be/[9]Be ratio of 2.85 x



$10^{-12}$ (Nishiizumi et al., 2007). Measured $1\sigma$ analytical uncertainty ranged from 1.77% to 3.43% (Table S1).
Apparent exposure ages were calculated using the online cosmogenic age calculator v3 (Balco et al., 2008) using the
Baffin Bay $^{10}$Be production rate calibration data set (Balco et al., 2008; Young et al., 2013). Apparent exposure age
refers to the calculated age if the samples were at the surface and experienced zero erosion. Although these apparent
ages are not used in our erosion models, they are instructive in analyzing and visualizing the context of the data
based on a priori assumptions.

**3.4 Cosmogenic Nuclide Modeling**
Following Balco et al. (2011), we created a 3D forward model (Graham, 2023) of cosmogenic $^{10}$Be production in the
upper 1.2 meters of the glacially eroded bedrock at Location A using the known exposure and burial history. The
history we adopt is shown in Fig. 4 and is as follows: Exposure from 7400 years ago to 1790 CE (~7200 years of
exposure), burial from 1790 to 2010 CE (220 years of burial/erosion), and exposure from 2011 to 2018 CE (year of
sample collection). We use the model to not only quantify the pre-quarrying surface, but also to determine the
sensitivity of the specific sampling locations in the resulting divot. We thus prioritized certain sample locations from
the vertical (lee) face to optimize the number of samples measured. To start, we simulated the $^{10}$Be concentrations
using a variety of pre-quarrying surface shape geometries ranging from a rectangular cross section to a triangular
cross section to a geometry that is the same as the present-day surface. End members of these pre-quarrying surface
options are illustrated as the purple, green and red lines in Figure 5B. Three-dimensional representations were
generated by extending the 2D surface profiles laterally. This simplified the hypothetical surface models and was
justified by the presence of laterally similar surface profiles observed on the landscape. Simulated cosmic particle
bombardment was prescribed based on Gosse and Phillips (2001) for azimuth and elevation angles through the
simulated overlaying bedrock to each sample location.

We next created an inverse model to solve for the pre-quarrying surface profile at Location A. An adaptive

Metropolis Hastings Markov Chain Monte Carlo (MCMC) Matlab solver package (Haario et al., 2006) was
implemented to estimate the parameters necessary to minimize the chi-squared reduction of the estimated $^{10}$Be
concentrations to the measured $^{10}$Be concentrations. The unknown parameters were: 1) the surface profile x
(horizontal distance within the quarried block) inflection point, 2) the surface profile z (depth) inflection point, 3)
the depth of surface abrasion applied equally across all samples, and 4) the absolute attenuation length ($\Lambda_{abs}$) of the
high energy neutron spallation through the rock. Acceptable a priori parameter ranges were initially prescribed
(Table 1). We used the MCMC inversion to solve for the posterior parameters that correspond to the minimized $^{10}$Be
concentrations through the chi squared reduction. Due to the relatively shallow maximum sample depth (~1.2
meters) and small amount of abrasion previously estimated by Balter-Kennedy et al. (2021), muon production is
minimal and approximately linear across the narrow depth range. Therefore, we treated production via muons as a
linear function of depth across all sample sites.

The surface profile was generated via a point with X,Z coordinates located within the pre-quarrying

geometries prescribed above. To expand laterally, a 25-point smoothed surface interpolation (Matlab function pchip)
was applied between the generated point and the edges of the quarried block (top of the stoss cliff, and the rough-to-



smooth transition around the perimeter of the quarried block). The initial estimate of abrasion depth for the model is
based on an abrasion depth estimate from the surface sample 18JAK-Surface following the methods described in
Briner and Swanson (1998) and Young et al. (2016) and is independent (but complementary) to results obtained by
Balter-Kennedy et al. (2021). The absolute attenuation length ($\Lambda_{abs}$) is based on the range of values estimated in
Gosse and Phillips (2001). Most estimations of spallation attenuation with depth rely on the apparent attenuation
length ($\Lambda_{app}$) because it assumes a horizontally infinite half space, or a flat surface profile, which the sample lays
beneath at some depth, z (cm). Due to the off-zenith incoming cosmic particles travelling through an increasing
length of mass, an integrated value of attenuation results in the apparent attenuation (Dunai, 2010). Because our
research incorporates a complex surface model, the absolute attenuation length is required to properly simulate the
attenuation through varying thicknesses of rock from off-zenith angles. Our inversion results in an estimate for an
absolute attenuation length of $184\pm13$ g cm$^{-2}$. When converted to an apparent attenuation length, via $\Lambda_{app} =$
$(3.3/4.3)*\Lambda_{abs}$, this becomes $141\pm10$ g cm$^{-2}$ and is within the range reported for the Arctic by Gosse and Phillips

(2001).


**3.5 Terrain analysis and volume of quarried material**
We applied the resulting most probable profile of the quarried block at Location A (see Results) to other divots that
were quarried during the historical advance. Incidentally, the shape of the quarried material is consistent with, and
could largely be defined by, the non-quarried surfaces surrounding the quarried divots. Informed by results from the
cosmogenic nuclide measurements at Location B and surface texture mapping, we identified which of the quarried
divots were excavated during the historical advance versus glacier overriding associated with the last glaciation. The
latter quarried zones were excluded from the analysis to prevent overestimating the quarried rock volume attributed
to the historical overriding event. All geographical information system (GIS) analysis was performed in QGIS
Desktop 3.16 Long Term Release, with all datasets transformed to NSIDC Sea Ice Polar Stereographic North. The
UAV-generated DEM, nominally 0.03 m raster cell size after transformation, was re-gridded to 0.05 m cell size to
which all further raster analysis was standardized to. We defined our field area based on the extent of an
exceptionally bedrock-rich part of the glacier forefield, with a higher degree of surface sediment cover around its
periphery. Some areas of sediment cover from within our outlined study zone are excluded because they occluded
accurate identification of the underlying surface texture and are not included in area calculations of the study site.

We defined the quarried zones attributed to the historical advance with polygons and removed them from

the DEM of the present-day surface. We then interpolated a synthetic surface across the missing holes in the DEM
to recreate the pre-1790 CE surface, or "paleo-surface" using the geometry guided by results from Location A. Next,
we generated a difference map between the paleo-surface DEM and the present-day surface DEM. We then summed
these values from the difference map. Finally, when applying the resultant abrasion rate across the study area, we
estimated a cavity area below each of the historically quarried zones (assuming a seasonally averaged cavity roof of
45°) and subtracted this area from the total study area.



## 4. Results

The $^{10}$Be concentrations from Location A (Table 2) decrease with depth and increase along the floor
outwards from the lee cliff base (Fig. 5B). All samples result in lower apparent exposure ages than the estimated
exposure duration of 7200 yr (7400 yr deglaciation minus 200 yr of subsequent burial), indicating that glacial
erosion recently took place. The best fit of our forward model is a triangular wedge shape of removed material (Fig.
5B & C, green). This shape is supported by the surface morphology and textures adjacent to the quarried divot.
Furthermore, this triangular wedge shape is supported by the MCMC model, which reveals a slightly concave pre-
quarried block surface (see "MCMC" in Fig. 5B). Additionally, our MCMC modelling using all samples at Location
A yielded a surface abrasion depth of 4.1±1.9 cm (Table 1). When using individual samples from the top (stoss) side
of the lee ledge and from beyond the quarried divot, we derive abrasion depths of 2.7±1.1 and 5.8±1.1 cm,
respectively.
To estimate an abraded volume of the study site, we consider several distinct abrasion rates calculated
across the study area. Combining abrasion depths mentioned above with four nearby values reported by Balter-
Kennedy et al. (2021) yields an average abrasion depth of 2.78±1.84 cm and an abrasion rate of 0.126±0.084 mm
yr$^{-1}$. Calculating the volume of material abraded requires the removal of areas where cavities existed in the
immediate lee of bedrock steps. Although cavities change in size seasonally, we estimate that 12% of the field area
consists of cavities assuming a 45° sloping cavity roof from the lip of bedrock steps. We thus calculate a volume of
322±204 m$^3$.
Results from Location B show significant differences in the measured $^{10}$Be concentrations between the two
lee steps. Sample ER2-A was collected at the base of an 85-cm-tall lee face that exhibits a fresh (non-polished)
surface texture and a darker color (Fig. 3C). Its apparent exposure age of 2.3 ka (accounting for shielding using
present topography) is significantly less than the expected age of ~7.2 ka, indicating quarrying during the historical
advance over the site. Sample ER2-B is from the base of a 120-cm-tall lee cliff and exhibits a lightly abraded texture
and lighter color (Fig. 3B). Its apparent exposure age when accounting for shielding using present topography is 6.9
ka. We attribute the difference in apparent age of sample ER2-B and its expected age of 7.2 ka to a few centimeters
of abrasion, and more importantly, to a lack of quarrying during the historical advance. Thus, the results from
Location B indicate that other bedrock steps that exhibit smoother, lightly abraded surfaces were quarried during the
prior glaciation, and that only rougher, darker-colored surfaces in some lee faces were quarried during the historical
advance.
Our field mapping of rock surface textures exhibits quarried zones with a mixture of rough and abraded lee
surfaces. We identified 73 quarried zones classified with rough-textured, dark-colored surfaces ("historical"
quarrying) and 84 quarried zones classified as having slight smoothing and lighter surface tone (quarrying during the
last glaciation; Fig. 6). We calculate an area of quarried material during the historical advance of 1,635–2,050 m$^2$
(the derivation of this range is discussed below) of the total 13,256 m$^2$ field area (12–15%) and a quarried volume of
378±45 m$^3$. Using the duration of overriding during the historical advance, this equates to an equivalent quarrying
rate of 0.13±0.03 mm yr$^{-1}$ when averaged across the study site. We calculate a combined (total) eroded rock volume



of 700±249 m$^3$ and total subglacial erosion rate of 0.26±0.16 mm yr$^{-1}$, of which 47% is attributed to abrasion and
53% is attributed to quarrying.

Measurements of ice flow indicators, including striations, crescentic gouges and chatter marks, reveal a

south (180°) to southwest (225°) ice-flow direction (Fig. 7). When sorted by type of ice flow indicator, a pattern
emerges showing an evolution of flow direction during the most recent ice advance. Small striations, being the most
likely to represent the final ice-flow direction before deglaciation, show the most recent ice flow direction toward
the south. Crescentic gouges, chatter marks and lee face orientations, which are more likely to persist after some
surface abrasion, reveal a southwesterly direction of ice flow. This shift likely represents the evolving flow direction
and velocity change as ice flow over the field area increased in velocity, shifted to the southwest and thickened
during the maximum phase of the historical advance. Based on the orientation of quarrying ledges and ice flow
indicators, it thus appears that much of the quarrying occurred when the ice flowed southwest during what was
presumably the highest ice flow velocity and thickness of the historical advance.

**5. Discussion**

We provide a new approach for quantifying the quarried volume of sediment across a glacial landscape and

for partitioning the relative contributions of quarrying and abrasion. Due to the inherent difficulty in measuring
quarrying directly, previous estimates rely on computational models or inferences made from measurements of
proglacial sediment discharge (Hallet, 1996; Loso et al., 2004; Riihimaki et al., 2005; Ugelvig et al., 2018).
Quarrying estimates from stream sediments (e.g., bedload) require assumptions about the portion of the suspended
load that is also derived from quarrying (Riihimaki et al., 2005). Here, our measurements of quarrying volume and
rate stem from the combination of in situ $^{10}$Be measurements and terrain analysis.

Our erosion rate measurements are similar to other estimates for glacial erosion in Greenland and beyond

(Koppes and Montgomery, 2009; Cook et al., 2020). Our total erosion rate of 0.32±0.09 mm yr$^{-1}$ is similar to what
Balter-Kennedy et al. (2021) found at the same site using both surface $^{10}$Be measurements (0.4–0.8 mm yr$^{-1}$) and a
$^{10}$Be depth profile from a 4-m-deep rock core (0.3–0.6 mm yr$^{-1}$). Although these rates are lower than those found
using a sediment-budget approach in southwestern Greenland (4.8±2.6 mm yr$^{-1}$; Cowton et al., 2012), they are
similar to centennial-scale erosion rate estimates of 0.29–0.34 mm yr$^{-1}$ in northwestern Greenland (Hogan et al.,

2020).

Quarrying is inferred to be highly dependent on glaciological and lithological conditions, including bedrock

hardness and fracture spacing (Dühnforth et al., 2010; Krabbendam and Glasser, 2011; Iverson, 2012). Our study
site contains competent, hard crystalline rock with widely spaced fractures (on the order of several meters). Thus,
based on these characteristics, we would expect abrasion to dominate at our field site (Anderson, 2014). However,
despite only 12–15% of the field site by area exhibiting recent quarrying, we calculate that 53% of total glacial
erosion occurred as quarrying.

Our MCMC results and field observations suggest that, prior to quarrying, the bedrock surface was

relatively low relief, likely with wave cavities in lee locations (Zoet et al., 2013) as opposed to stepped geometries
that are more often considered in theoretical studies of quarrying (e.g., Anderson et al., 1982; Hallet, 1996; Iverson,



2012; Anderson, 2014). Despite bedrock characteristics inhibiting quarrying, the Greenland Ice Sheet experiences
significant seasonal and sub-seasonal changes in subglacial hydrology in this area (Das et al., 2008; Andrews et al.,
2014), which is thought to aid quarrying processes (Anderson, 2014; Ugelvig et al., 2018). Propagating fractures that
are presumed to eventually lead to failure and quarrying appear to not solely rely on pre-existing fractures in the
bedrock at our study site, but could have been induced from processes related to the formation of crescentic gouges
(Gilbert, 1905) that we observed in abundance in the field (Fig. 2A). That many crescentic gouge trains increase in
size toward quarried ledges–with a crescentic gouge at the lip of many edges–may indicate that gouge formation is a
fracture nucleation point that leads to quarrying events in this field area (Figs. 2 and 8).

There are uncertainties associated with calculating erosion depth, volume and rate. We do not expect

uniform abrasion across the study area given the stepped nature of the terrain and localized variations in basal stress.
At Site A, we find a lower abrasion depth at the lip of the divot (2.7±1.1 cm) than the floor beyond the quarried zone
(5.8±1.0 cm). We do not have enough data to elucidate predictable spatial patterns of more or less abrasion across
the study site; instead, we rely on an average of a number of data points that provide a useful representative abrasion
depth to apply across our field area.

It is useful to further consider uncertainties, such as those perhaps associated with our erosion thickness,

volume and rate results. Abrasion depth estimates reported here have high uncertainty due to the inherent
measurement error in measuring cosmogenic nuclide concentration. An analysis of errors in Young et al. (2016) and
Balter-Kennedy et al. (2021) for shallow abrasion depths shows a consistently appreciable uncertainty in relation to
the low magnitudes of rock removal via abrasion. The measurement uncertainties for samples in the companion
study of Balter-Kennedy et al. (2021) is ~2.5–3 cm, but when the estimated depth of abrasion is small and similar to
mean uncertainty, the uncertainty can result in a significant range of the abraded depth. One advantage of our
experiment at Location A is that multiple samples were used in the MCMC inversion, reducing the uncertainty in the
estimated abrasion depth. Unfortunately, even with the added resolving power of multiple samples, the uncertainty
in the abrasion depth is still 46%.

When converting the abrasion depth to an abrasion rate, another source of uncertainty is the duration of

erosion. Whereas the timing of recent deglaciation and exposure is well constrained, the timing of burial is less well
constrained. We use the overriding duration of 1790–2010 CE used in Balter-Kennedy et al. (2021), which is based
on prior work in the area (Briner et al., 2011; Young et al., 2016). Although we use an absolute date range in our
erosion rate calculations, the initiation glaciation at the onset of the historical advance at our study site is
reconstructed, not observed, and the initiation timing of overriding would affect the calculated erosion rates. If the
ice arrived decades earlier (we think this is more likely than ice arriving later than 1800 CE), our calculated erosion
rates would decrease, but the ratio of abrasion to quarrying would be unaffected.

An additional source of uncertainty relates to the reconstructed profile of the paleo-surface slope of

quarried blocks, and thus of the volume of each removed block. We use the three-dimensional nuclide production
inversion of the quarried zone at Location A to guide the shape for other quarried zones. To estimate the uncertainty
of each quarried block, and the cumulative uncertainty of the quarried volume across the study site, each zone was
analyzed for the likelihood of having a pre-quarrying sloped, triangular profile versus a more rectangular, stair-step



profile. Of the 73 quarried zones, 63 were identified as triangular shape based on the localized topography around
each quarried zone, as was the case at Location A that we confirmed with our cosmogenic nuclide measurements
and modeling. The remaining 10 locations were identified as likely to have been rectangular blocks, and the rock
volume quarried at these sites was calculated by doubling the volume generated by a triangular cross-section.
The uncertainty in our estimates of quarried rock volume is independent of the cosmogenic nuclide
concentration. To estimate uncertainty in our manual outlining of each area of the quarried zones, a 0.5-meter buffer
was extended at the edge of the floor of each quarried zone; this edge is based on changes in surface texture from
rough to smooth as recorded in the high-resolution orthoimagery. The location of this transition is also dictated by
the presence/absence of chatter marks/crescentic gouges, surface patina, and rock color. While many locations have
a well-defined transition, 0.5 meters is an upper limit on our ability to define this boundary. The lee cliff is a well-
defined feature on the landscape, and is accurately identified from the orthoimagery, with assistance using other
products such as the DEM, and Hillshade/Roughness QGIS processing products. We consider our 0.5 m buffer on
the extent along the quarried floor to be a conservative estimate. When used to define the volume of each block, we
find that the 0.5 m buffer equates to a volume range of $379 \pm 45$ m$^3$, and a quarried area of $1842 \pm 100$ m$^2$ (12–15% of
the study area).
Our inverse modeling of cosmogenic nuclide production at Location A highlights the continued importance
of cosmogenic nuclides in glacier erosion studies. Optimizing sampling locations to estimate the parameters of
interest (surface geometry of a removed block, depth of abrasion, and attenuation length) was important for our
inversion results. The sensitivity analysis to determine how samples were important in our forward model scenarios
aided in sample selection for processing. The samples along the horizontal lee floor (FL1, FL2) are the most
important for constraining the surface profile shape. Samples at the present-day surfaces (Surface, FL3) are the most
important for constraining the depth of abrasion, while the samples collected along the vertical lee cliff are sensitive
to the depth of abrasion and the attenuation length. In fact, not all samples collected along the vertical cliff were
needed for the analysis, while additional samples along the floor near the quarrying-abrasion transition could have
been beneficial.
**6. Conclusion**
Our pairing of cosmogenic nuclide analysis with inverse modeling of cosmogenic nuclide
production through quarried material, along with topographic and morphologic analysis of a recently deglaciated
bedrock landscape, provides one of the first direct observation-based estimates of glacial quarrying and partitioning
of glacial erosion processes. We found that quarried volume generally matched that of abrasion despite a hard
crystalline bedrock with wide fracture spacing and a low-relief surface morphology, all conspiring to limit
quarrying. It seems that quarrying mostly took place via triangular wedge removal at this site. Field observations
suggest clast-bed impacts evidenced by abundant crescentic gouges are a possible mechanism to nucleate quarrying
events, assisted by seasonal and sub-seasonal fluctuations in subglacial water pressure. These results are a small
addition to a field that needs further analysis. Yet, field data like these are important for grounding landscape
evolution models with observational datasets and for providing fundamental information for understanding coupled



glacier-hydrology-sediment production processes. Ultimately, the results of our work invite further analysis at this
field site, including testing of both theoretical and computational models of glacial erosion.

**Code and data availability**
Code and data are available on GitHub at https://github.com/w0gpr/Cosmo3D (last access: April 26, 2023) and
Zenodo (https://doi.org/10.5281/ZENODO.7858913; Graham, 2023).

**Author contribution**
BG, JPB, NEY and AB-K designed the study and collected field data. BLG, JPB and JMS led rock sample
preparation and [10]Be analysis. BG modeled [10]Be production, computed terrain analysis, and derived erosion results.
MK, KP and EKT provided significant input throughout the course of this research. BG and JPB prepared the paper
with contributions from all co-authors.

**Competing Interests**
The contact author declares that none of the authors have any competing interests. Kristin Poinar is a member of the
editorial board of The Cryosophere

**Acknowledgements**
We thank Chris Sbarra and Rosanne Schwartz for sample processing, CH2MHill Polar Field Services for supporting
fieldwork, and Alan Hidy at Lawrence Livermore National Laboratory for beryllium isotope measurements.

**Financial support**
This research was supported by US National Science Foundation award #1504267 to Briner and #1503959 to Young
and Schaefer.

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

**Figure captions**

**Figure 1.** A. Greenland. B. Jakobshavn region Aug 2018; extent of Jakobshavn Isbræ (J.I.) in 1850 CE;
J.If.=Jakobshavn Isfjord; nb/sb=north branch/south branch. C. Study area showing glacial erosion depths from
Balter-Kennedy et al. (2021) and this study (star). D. Oblique drone photograph of the study area (point of view
shown in C) showing study site A and B.

**Figure 2.** Photographs of surfaces in the study area. A) Heavily abraded and polished surface showing one of the
many "gouge trains;" view to SW. B) Small lee step (approximately 20 cm high) within a heavily abraded and
polished zone; note downflow from the lee cliff is a zone with more lightly abraded surfaces. C) Fresh surfaces with
minor grain-to-grain relief and limited evidence for abrasion shown within quarrying 'scars.' D) Focus on a lee step
(approximately 1 m high) showing the transition from a heavily abraded stoss surface (lightly colored) to darker-
colored, fresh lee faces; some of the dark color in this image is from subglacial precipitate "staining."

**Figure 3.** A) Study location A; blue area is extent of the quarried material. Stars are locations of $^{10}$Be measurements.
B) Study location B; pair of quarried zones with a fresh, rough lee surface (left; sample ER2-A), and smooth,
abraded lee surface (right, sample ER2-B).

**Figure 4.** Concept model for $^{10}$Be production and concentration for the field area. 1) Retreat of the ice sheet from
the field area 7.4 ka. Erosion during the last glaciation is sufficient to remove $^{10}$Be to background levels. 2) The
paleo-surface is exposed to cosmic radiation during the Holocene until ice overrides at ~1790 CE, building up $^{10}$Be
in the upper ~2 m of bedrock. 3) Ice readvances and erodes via abrasion and quarrying during the historical
advance. 4) The present-day surface is exposed in 2010 CE.


**Figure 5.** A) Photograph of Location A (see also Fig. 3A) showing fresh quarried face and floor. B) Cross section
representation of the 3D model domain for Location A. Sample locations are marked as black boxes. The red line
shows the present-day surface profile, while purple and green lines show rectangular and triangular pre-quarrying
surface profiles, respectively, used in forward model. The thin gray lines are the minimized surface profiles from the



MCMC inversion. C) Measured (small circles) and simulated (lines in color) $^{10}$Be concentration of the three forward
model scenarios; colors match top.

**Figure 6.** A. Orthoimage of the field area showing fractures (blue lines) and lee cliff faces (red lines). Zones
quarried during the most recent glacial advance are outlined in purple. Rose diagram (inset) shows all measured ice
flow indicators (in the direction of ice flow). B. Elevation difference in quarried divots assigned to block removal
during the historical advance.

**Figure 7.** The orientation of ice-flow indicators subdivided into type. Blue lines encompass orientations from all
ice-flow indicators combined (see Fig. 6).

**Figure 8.** Photographs showing the relationship between crescentic gouges and quarrying at the field site. A) Gouge
trains leading to a lee face with evidence for quarried flakes initiated by a gouge process (ice flow from upper right
to lower left). B) Example of angled (and polished) lee face from which a relatively thin flake has been quarried and
removed.





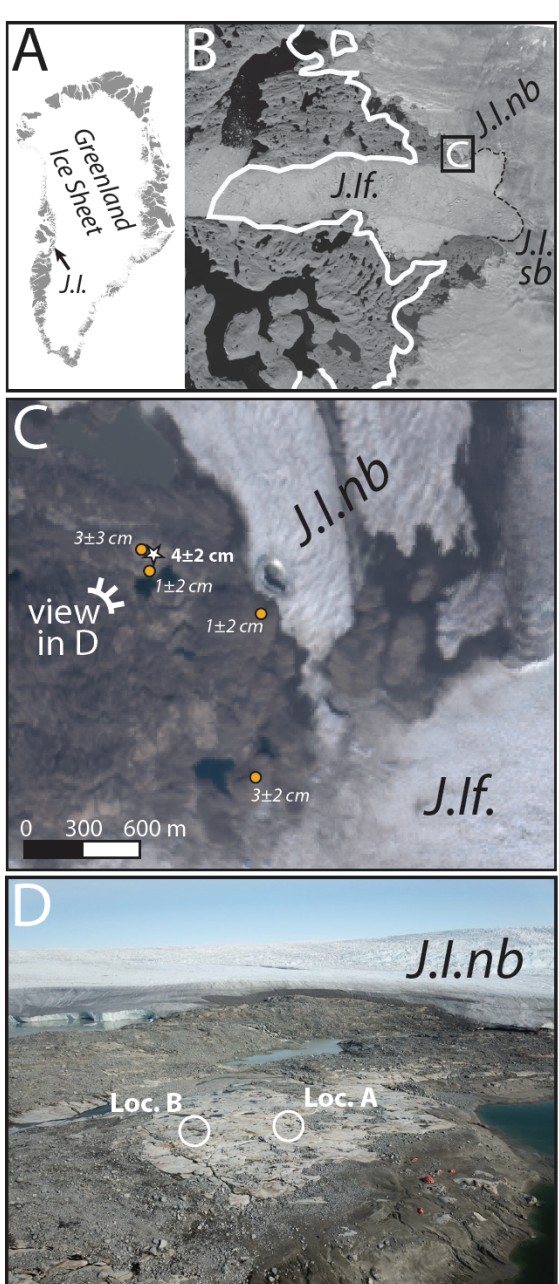

**Figure 1.** A. Greenland. B. Jakobshavn region Aug 2018; extent of Jakobshavn Isbræ (J.I.) in 1850 CE; J.If.=Jakobshavn Isfjord; nb/sb=north branch/south branch. C. Study area showing glacial erosion depths from Balter-Kennedy et al. (2021) and this study (star). D. Oblique drone photograph of the study area (point of view shown in C) showing study site A and B.



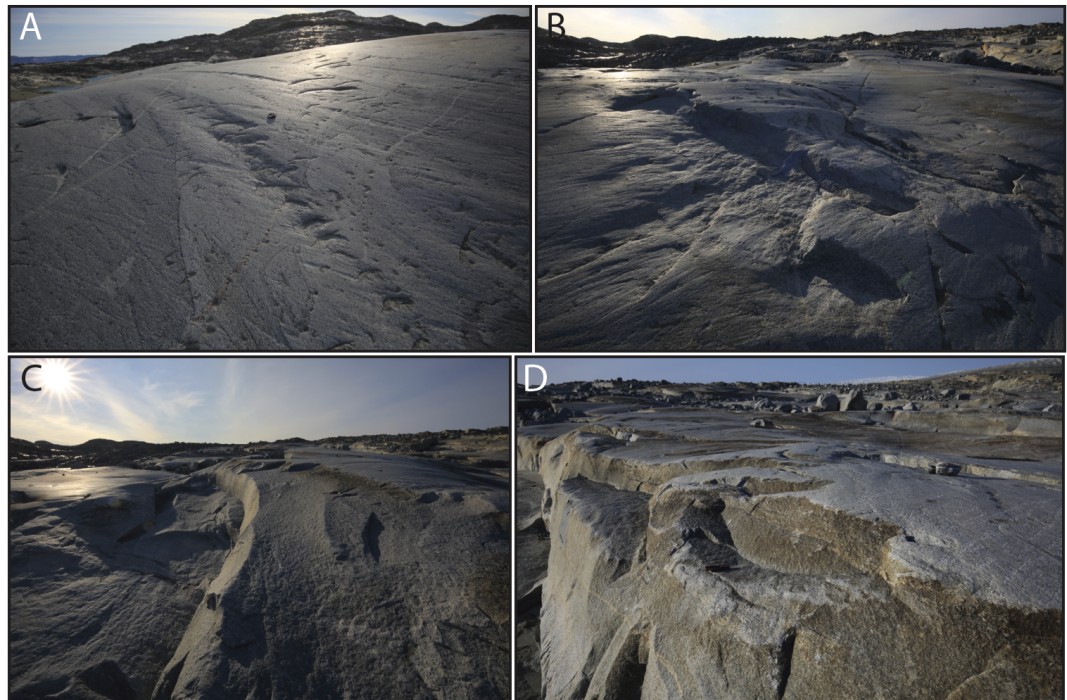

**Figure 2.** Photographs of surfaces in the study area. A) Heavily abraded and polished surface showing one of the many "gouge trains;" view to SW. B) Small lee step (approximately 20 cm high) within a heavily abraded and polished zone; note downflow from the lee cliff is a zone with more lightly abraded surfaces. C) Fresh surfaces with minor grain-to-grain relief and limited evidence for abrasion shown within quarrying 'scars.' D) Focus on a lee step (approximately 1 m high) showing the transition from a heavily abraded stoss surface (lightly colored) to darker-colored, fresh lee faces; some of the dark color in this image is from subglacial precipitate "staining."





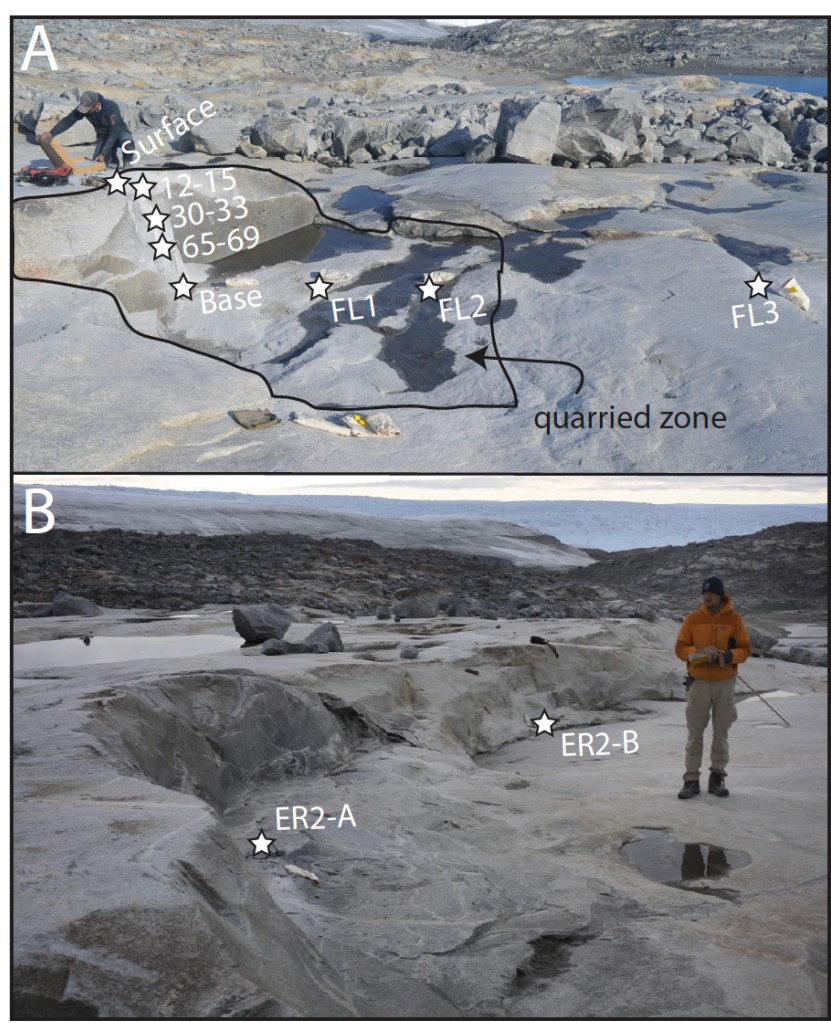

**Figure 3.** A) Study location A; blue area is extent of the quarried material. Stars are locations of $^{10}$Be measurements.
B) Study location B; pair of quarried zones with a fresh, rough lee surface (left; sample ER2-A), and smooth,
abraded lee surface (right, sample ER2-B).



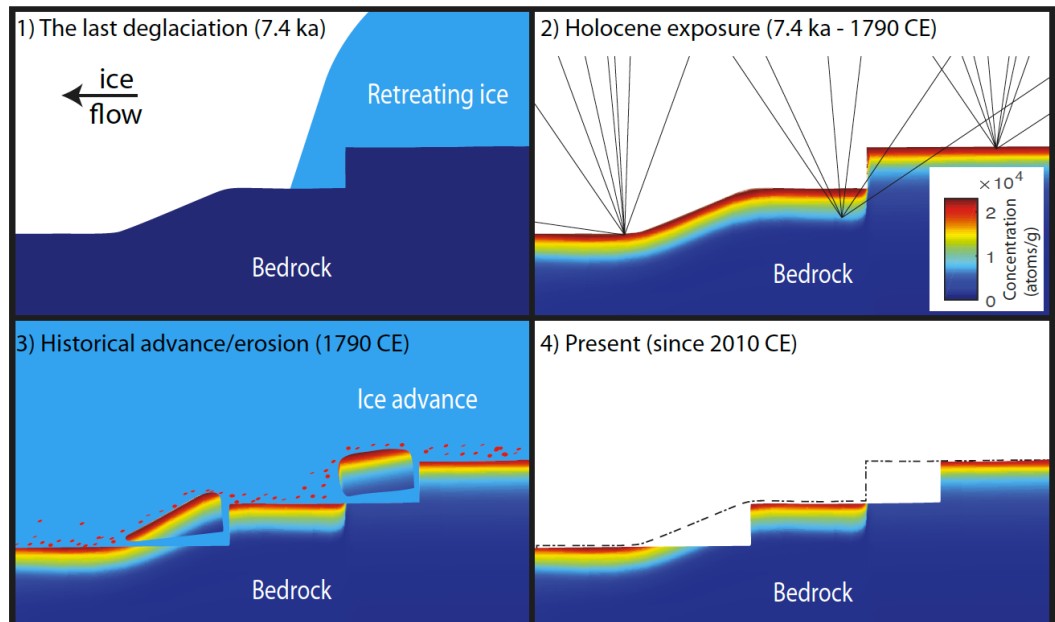

**Figure 4.** Concept model for [10]Be production and concentration for the field area. 1) Retreat of the ice sheet from the field area 7.4 ka. Erosion during the last glaciation is sufficient to remove [10]Be to background levels. 2) The paleo-surface is exposed to cosmic radiation during the Holocene until ice overrides at ~1790 CE, building up [10]Be in the upper ~2 m of bedrock. 3) Ice readvances and erodes via abrasion and quarrying during the historical advance. 4) The present-day surface is exposed in 2010 CE.



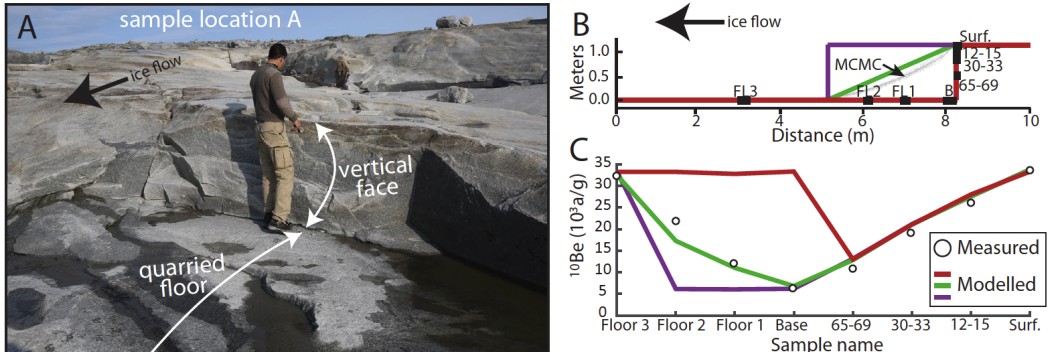

**Figure 5.** A) Photograph of Location A (see also Fig. 3A) showing fresh quarried face and floor. B) Cross section representation of the 3D model domain for Location A. Sample locations are marked as black boxes. The red line shows the present-day surface profile, while purple and green lines show rectangular and triangular pre-quarrying surface profiles, respectively, used in forward model. The thin gray lines are the minimized surface profiles from the MCMC inversion. C) Measured (small circles) and simulated (lines in color) $^{10}$Be concentration of the three forward model scenarios; colors match top.



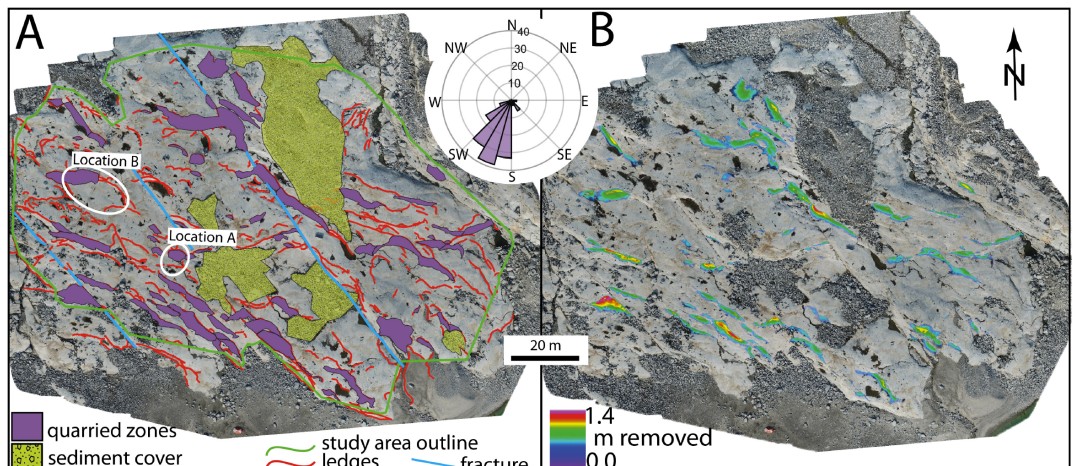

**Figure 6.** A. Orthoimage of the field area showing fractures (blue lines) and lee cliff faces (red lines). Zones quarried during the most recent glacial advance are outlined in purple. Rose diagram (inset) shows all measured ice flow indicators (in the direction of ice flow). B. Elevation difference in quarried divots assigned to block removal during the historical advance.



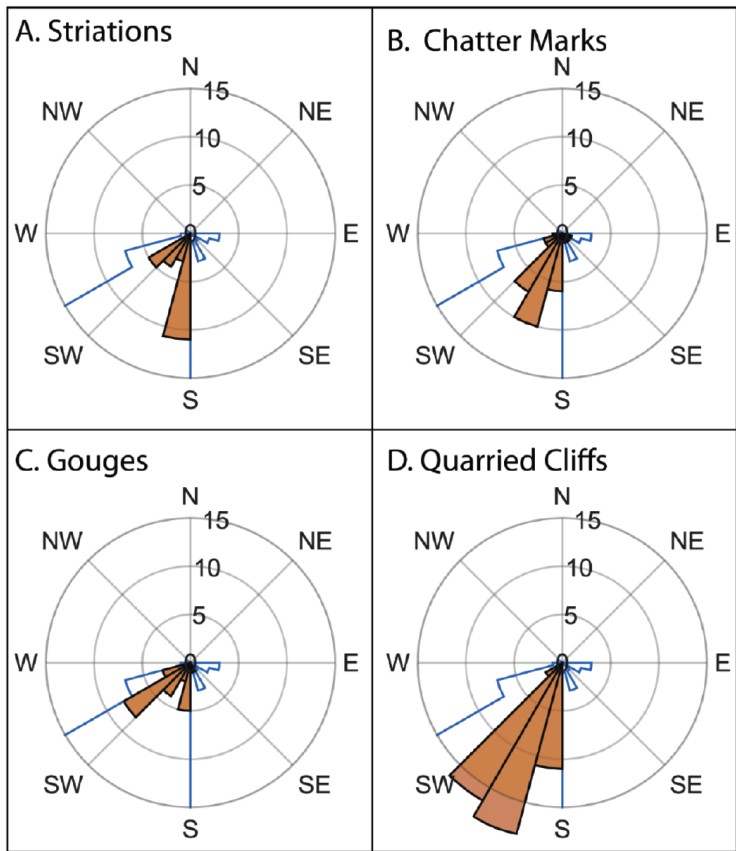

**Figure 7.** The orientation of ice-flow indicators subdivided into type. Blue lines encompass orientations from all ice-flow indicators combined (see Fig. 6).



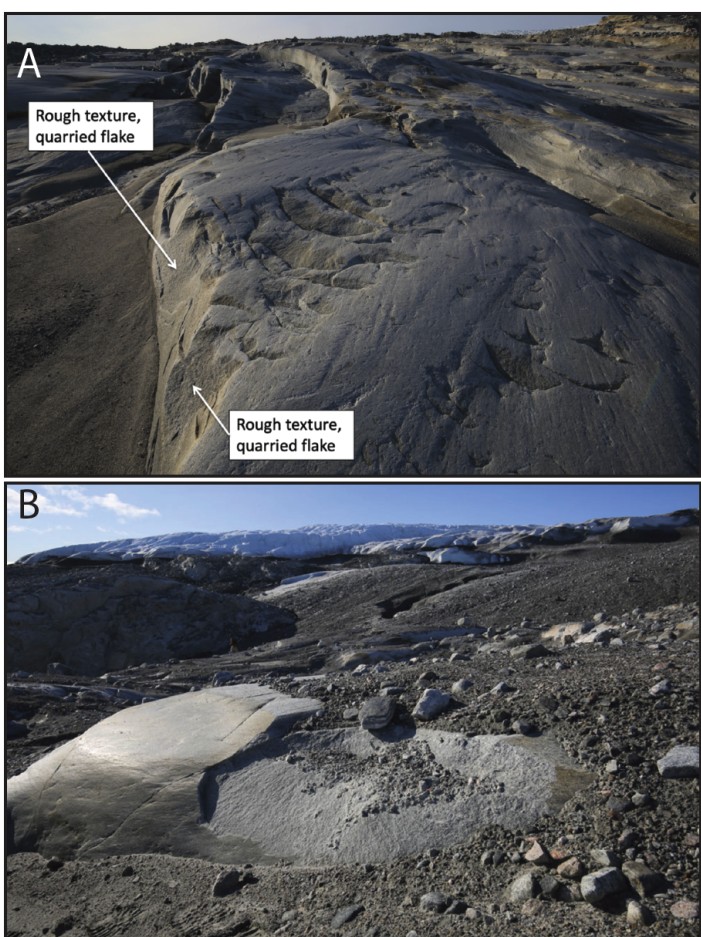

**Figure 8.** Photographs showing the relationship between crescentic gouges and quarrying at the field site. A) Gouge trains leading to a lee face with evidence for quarried flakes initiated by a gouge process (ice flow from upper right to lower left). B) Example of angled (and polished) lee face from which a relatively thin flake has been quarried and removed.



**Table 1.** MCMC parameters *a prior* and posterior.

| Parameter Name | Input | | | Output | |
| --- | --- | --- | --- | --- | --- |
| | Initial Guess | Minimum | Maximum | mean | std |
| xPoint | 0.6 | 0 | 1 | 0.70353 | 0.18268 |
| zPoint | 0.6 | 0 | 1 | 0.50464 | 0.22515 |
| Lambda (g cm$^{-2}$) | 208 | 150 | 240 | 184.26 | 12.518 |
| Abrasion Depth (cm) | 2.75 | 0 | 10 | 4.1375 | 1.9038 |

The *a priori* input into the MCMC inverse and the posterior output from the model runs that minimized the chi squared reduction.

**Table 2.** Beryllium-10 sample data.

| Sample Name | Qtz weight (g) | Carrier added (g) | $^{10}Be/^{9}Be$ (x10$^{-14}$) | Blank corrected $^{10}Be$ (atoms g$^1$) | Apparent Age (yr) |
| --- | --- | --- | --- | --- | --- |
| Surface | 56.821 | 0.1817 | 15.3±0.3 | 33,610±630 | 7,100±130 |
| 12-15 | 55.872 | 0.1822 | 11.7±0.2 | 26,090±490 | 5,510±100 |
| 30-33 | 60.139 | 0.1830 | 9.2±0.2 | 19,130±360 | 4,030±80 |
| 65-69 | 64.043 | 0.1818 | 5.5±0.1 | 10,730±240 | 2,260±50 |
| Base | 34.677 | 0.1820 | 1.8±0.1 | 6,210±210 | 1,310±40 |
| FL1 | 34.663 | 0.1832 | 3.3±0.1 | 11,910±320 | 2,510±70 |
| FL2 | 45.816 | 0.1835 | 8.0±0.2 | 21,870±500 | 4,630±110 |
| FL3 | 33.552 | 0.1832 | 8.5±0.2 | 31,940±570 | 6,740±120 |
| ER2-A | 26.190 | 0.1832 | 2.0±0.1 | 9,150±310 | 1,930±70 |
| ER2-B | 73.360 | 0.1834 | 14.6±0.3 | 25,090±480 | 5,300±100 |

Location A samples were located at 69.2316°N and 49.8093°W, and Location B samples (ER2) at 69.2318°N and 49.8103°W. All samples were at an elevation of 107 meters above sea level. Sample density was 2.65 g cm$^{-3}$, and the 07KNSTD Be standard was used. The Apparent Age is the St scaling apparent exposure duration, assuming no shielding.

**Table 3.** Eroded rock volume and glacial erosion rates

| | Volume (m$^3$) | Area. (m$^2$) | Rate (mm yr$^{-1}$) |
| --- | --- | --- | --- |
| Abrasion | 323±204 | 11,623* | 0.13±0.08 |
| Quarrying | 378±45 | 1843±208 | 0.13±0.03 |
| Total | 700±249 | 13,256 | 0.26±0.16 |

*value is 12% less than total area because of estimated area of subglacial cavities