# Peer review of "In situ 10Be modeling and terrain analysis constrain subglacial"

_EGUsphere, 2023_

## Referee Comment (RC1)

**Review of**

**Graham et al.**

*In Situ 10Be modelling and terrain analysis constrain quarrying and abrasion at Jakboshavn Isbrae, Greenland.*

This is a great study, and very important to finally quantify what has been for > 100 yrs regarded as the main subglacial erosion mechanisms. The study design is sound, the results convincing and generally well presented, and limitations and uncertainties well discussed. I am not familiar with the details of cosmogenic dating methods as such, and I trust that another reviewer will comment on this.

This Manuscript should be published. As always, however, a manuscript can always be improved, so here goes. Some minor re-organisation is needed, some extra background information in Setting, and some other minor improvements are suggested.

**General comments**

1. It took me while that we're really here looking at a mini-glaciation, or a advance-retreat cycle of c. 200 yr duration. This should be more clearly mentioned in the Abstract.
2. Reporting of Results. The figures of eroded volume are really quite meaningless, as this is only valid for a specific area. It would be much clearer and meaningful to report erosion depth (and erosion rate, as is done already). Thus: "depth of X mm by abrasion and Y mm of quarrying, averaged over a study area of 1.33 kmsq".
3. The only section that is weak is the **Study Area**. This needs a fair bit of work. Extra info, which can help to place the Results in context, are missing.
   a. The whole section is unclear to me – not being familiar with the Jakobshavn setting. (And given the general importance of this MS, hopefully many other readers are similar). It needs to be made much clear that this concerns a 'mini-glaciation', or a 220 yr advance-retreat cycle in historic times, not the main Last Glacial Maximum.
   b. The provenance or justification of the advance date should be dealt with here, not in the Discussion. Why not 1750 CE, or 1700 CE or 1600CE? It is also a bit odd to have a poorly constrained start date '(~1790) is then used to duration with a narrowly defined error (220 +- 5 yr).
   c. The rock type should be mentioned here. It's not mudstone or sandstone, right? (It looks like some granodioritie / tonalitic gneiss (?), but surely there's a bedrock geology map that can be consulted? (See also Comment Line 262 below).
   d. Surely something is known about the (maximum) thickness of the ice at the study site, if only from earlier topographic maps or early satellite imagery?
   e. Equality, is something known from satellite or other observations in the 1980s to 2000s about the ice velocity?
   f. Line 75-77 should move to Study Setting, not in Methods.
4. Some geomorphological results are only mentioned in the Discussion (line 306-309). These are Results, and should move to that section – not presented so late in the Manuscript.

**Specific Comments**

**Lines**

47-48. I guess would be fair to emphasise that that the sediment flux measurements are *indirect* proxies, and not direct measurements. (And maybe later in the discussion that such measurements can only work for small ice catchments, as in large ice sheets the grains size gets smaller with longer subglacial fluvial transport distances & would over-estimate the Abrasion component……).

98. Mention the distances from the quarried step/face. ".. two from the quarried floor, 1 and 2 m from the quarried step/edge, and one from a polished surface more distal (4m) from the quarried step/edge..".

208. 'four nearby results'. Do these occur on Figure 6? If so, their position should be plotted on that figure. Or on Figure 1. Also, we get no idea of the variability of the total of five abrasion depth values: this should be added – best as a separate, small table.

210-213. If reporting only averaged abrasion depth and not volume (see General Comment 2), these lines can go, but a calculation of the average abrasion depth should be written instead. That would be more informative, and less awkward.

238. Lee face orientation is a poor constraint on ice-flow direction, as it is commonly heavily predicated or influenced by the orientation of the pre-existing fractures (joints), and can thus deviate by up to 40° from the ice-flow direction; see study of Hooyer, et al. (2012). *Control of glacial quarrying by bedrock joints*. Geomorphology, 153, 91-101. I strongly suggest to take this out.

273. "… processes related to the formation of crescentic gouges (Gilbert 1905)…".

   1. Should be Gilbert 1906??

   2. Although Gilbert (1906) is a great paper, well ahead of its time, there are quite a few more modern relevant references: Harris, 1943; Embleton & King, 1975; Ficker et al., 1980; Wintges, 1985; Prest 1983; Drewry, 1986; Glasser and Bennett, 2004, and Krabbendam et al. 2009. The latter gives a reasonably good overview of the terminology, too (although dealing with a special case)

   3. The phrasing here is very vague. I think it would be good and informative to mention the *actual* process, namely exertion high clast-bed contact forces, exerted by large boulders embedded in basal ice pressing onto the bed. (see also Comment on Line 339).

297-298. It is still unclear to me what the justification for 1790 CE is for the start date of the historic glaciation – and not 1750 or 1700… A short sentence, summarising the previous work would be helpful here.

262. hard crystalline rock, fracture spacing .. AAAH! This should be described in the Setting!

339. This is the first mention of clast-bed forces, which reads very odd to people not familiar with crescentic gouges. This is why this should be explained near Line 273.

**Minor text issues (mainly to clarify things).**

Underline means suggested addition to text.

**Title:** why not mention 'rates'. "In Situ 10Be modelling and terrain analysis constrain quarrying and abrasion ***rates*** at Jakboshavn Isbrae, Greenland". Just to set it apart from the more general geomorphological papers around….

**Lines**

36. "… rock fragments are entrained in basal ice and pressed..". Clarity..

51. ".. a well constrained, short-duration (c. 200 yr) advance-retreat cycle…"

78. possibly mention surface roughness? Mm-scale roughness, or similar. " noted rock surface texture and roughness to distinguish abraded from quarried surfaces…" or similar.

83. ".. divot…" this seems to be mainly a rather obscure term originating from golf….? Would "hollow" or similar not be better..? I feel this either needs changing throughout, or explained at first use. Throughout the manuscript…

84. ".. a sharp transition from rough fractured to smooth abraded surface texture.."

86. ".. to estimate the chape of the quarried volume (single… "

89-90. This is awkward phrased: try to improve.

93-95. Bit of repetition here. Most of Line 93-94 are repeated in next sentence. Why not: "At location A, we sampled and measured 10Be concentrations in five samples…."

111. "….(1: freshly fractured  surfaces…."

112: "…. that the fresh-appearing fracture surfaces …"

114. first use of gouges can be deleted.

200. "..erosion took place recently."

222-224. Great! I like this!

223. "… prior main glaciation" (or some other word).

225. ".. a mixture of rough fractured and smooth abraded lee surfaces."

247. "… for establishing the relative contribution.." (Suggestion only)

248. ".. on computational models or proxy inferences made from …." For emphasis…

266-269. I like this!

300. "… but the ratio of abrasion to quarrying and the total depth of glacial erosion over the last historic glaciation would be unaffected."

Good luck with this – I truly hope this gets published. You guys actually scooped us with these methods - hohum, but such is life.

Maarten Krabbendam, Edinburgh, 25 May 2023

---

## Author Response (AR1)

**Response to Reviewer Comments**
Reviewer comments in black text, **our response in bold text**

**Reviewer 1**
Review of Graham et al. *In Situ 10Be modelling and terrain analysis constrain quarrying and abrasion at Jakboshavn Isbrae, Greenland*.

This is a great study, and very important to finally quantify what has been for > 100 yrs regarded as the main subglacial erosion mechanisms. The study design is sound, the results convincing and generally well presented, and limitations and uncertainties well discussed. I am not familiar with the details of cosmogenic dating methods as such, and I trust that another reviewer will comment on this.

This Manuscript should be published. As always, however, a manuscript can always be improved, so here goes. Some minor re-organisation is needed, some extra background information in Setting, and some other minor improvements are suggested.

**Firstly, we thank the reviewer very much for taking the time for a careful read and providing very sound suggestions for improvement!**

General comments
1. It took me while that we're really here looking at a mini-glaciation, or a advance-retreat cycle of c. 200 yr duration. This should be more clearly mentioned in the Abstract.
**Now we can write (added words in italic): "…we quantify subglacial erosion rates and constrain the ratio of quarrying to abrasion during a *recent, ~200-year-duration* overriding of a bedrock surface fronting Jakobshavn Isbræ, Greenland…"**

**2.** Reporting of Results. The figures of eroded volume are really quite meaningless, as this is only valid for a specific area. It would be much clearer and meaningful to report erosion depth (and erosion rate, as is done already). Thus: "depth of X mm by abrasion and Y mm of quarrying, averaged over a study area of 1.33 kmsq".
**We will work to make figures more clear. At present we do not have a figure showing eroded volume; Figure 6b shows erosion depth at the quarried sites. We hesitate to create a depth of quarrying averaged over the study area as that implies a uniform depth of erosion by quarrying. Abrasion, as a process, is likely to be closer to uniform; quarrying of course is highly localized.**

**3.** The only section that is weak is the **Study Area**. This needs a fair bit of work. Extra info, which can help to place the Results in context, are missing.
**See point by point response below.**

a. The whole section is unclear to me – not being familiar with the Jakobshavn setting. (And given the general importance of this MS, hopefully many other readers are similar). It needs to

be made much clear that this concerns a 'mini-glaciation', or a 220 yr advance-retreat cycle in historic times, not the main Last Glacial Maximum.

**Will add more text with the broader ice sheet history background. Hopefully, readers will find this useful, and see more clearly that this paper focuses on the Little Ice Age advance/retreat cycle and not the LGM.**

**"Greenland Ice Sheet margins are presently retreating, exposing terrain that was ice-covered during the latest Holocene advance that generally coincides with the Little Ice Age (Kjær et al., 2022). The north and south branches of Sermeq Kujalleq merged and extended westward ~35 km within the fjord to attain the "historical limit," which was observed in the fjord in 1850 CE (Fig. 1; Weidick and Bennike, 2007). Along the fjord, the historical limit is represented by a recognizable trimlines, and north and south of the fjord, prominent end moraines can be mapped to demarcate the extent of the "historical advance." In addition to the 1850 CE observation, this latest Holocene advance and retreat cycle has been dated in this region with lake sediment records (Briner et al., 2010, 2011) and a variety of imagery datasets (Csatho et al., 2007). The retreat of ice at our study site took place between 2008 and 2010.**

**During the Last Glacial Maximum, the Greenland Ice Sheet margin in the Sermeq Kujalleq sector rested on the continental shelf edge in Baffin Bay far west of Disk Bugt (e.g., Hogan et al., 2016). During the last deglaciation, the ice-sheet margin retreated eastward and eventually onto land on the eastern shores of Disko Bugt around 10,000 years ago. Later the ice margin retreated to within (east of) the extent of ice later attained during the historical limit."**

b. The provenance or justification of the advance date should be dealt with here, not in the Discussion. Why not 1750 CE, or 1700 CE or 1600CE? It is also a bit odd to have a poorly constrained start date '(~1790) is then used to duration with a narrowly defined error (220 +- 5 yr)."

**Great, thank you. This now addressed below – see our reply to comment #2 made by the second reviewer.**

c. The rock type should be mentioned here. It's not mudstone or sandstone, right? (It looks like some granodioritie / tonalitic gneiss (?), but surely there's a bedrock geology map that can be consulted? (See also Comment Line 262 below).
**Rock type is mentioned: "…to quantify total subglacial erosion rates of the gneissic bedrock in this area."**

d. Surely something is known about the (maximum) thickness of the ice at the study site, if only from earlier topographic maps or early satellite imagery?
**The maximum thickness of the ice over the study area during the LIA is not known, since this time period (1850 or earlier) is prior to aerial or satellite imagery.**

e. Equality, is something known from satellite or other observations in the 1980s to 2000s about the ice velocity?

**The Howat (2020) Greenland ice velocity product has derived local ice velocities from Landsat 4/5 in 1985–86. The image correlation over the slow-moving ice here is highly imperfect, though, with ice flow direction spanning a wide range of the compass rose (flowing to the north, east, and south in different image pairs). This agrees poorly with our more reliable geologic observations of ice flow to the southwest (Figure 7). Therefore, we are reluctant to report with confidence precise ice flow measurements here. They seem to be roughly 150–300 m/yr in spring/summer 1985–86, which is when observations are posted.**

**We add: "Although the velocity of ice over the study site during the maximum phase of the advance is not known, in 1985 when the site was still covered, surface velocity is in the 150-300 m yr$^{-1}$ range (Howat, 2020)."**

**Howat, I. (2020). MEaSUREs Greenland Ice Velocity: Selected Glacier Site Velocity Maps from Optical Images, Version 3 [Data Set]. Boulder, Colorado USA. NASA National Snow and Ice Data Center Distributed Active Archive Center. https://doi.org/10.5067/RRFY5IW94X5W. Date Accessed 07-27-2023.**

f. Line 75-77 should move to Study Setting, not in Methods.
**Done.**

**4.** Some geomorphological results are only mentioned in the Discussion (line 306-309). These are Results, and should move to that section – not presented so late in the Manuscript.
**This is now a paragraph in the results section; the text we moved from the Discussion to the Results section is noted in italics.**

**"Our field mapping of rock surface textures exhibits quarried zones with a mixture of rough and abraded lee surfaces. We identified 73 quarried zones classified with rough-textured, dark-colored surfaces ("historical" quarrying) and 84 quarried zones classified as having slight smoothing and lighter surface tone (quarrying during the last glaciation; Fig. 6). *Of the 73 quarried zones, 63 were identified as triangular shape based on the localized topography around each quarried zone, as was the case at Location A that we confirmed with cosmogenic nuclide measurements and modeling. The remaining 10 locations were identified as likely to have been rectangular blocks, and the rock volume quarried at these sites was calculated by doubling the volume generated by a triangular cross-section."***

Specific Comments
Lines
47-48. I guess would be fair to emphasise that that the sediment flux measurements are indirect proxies, and not direct measurements. (And maybe later in the discussion that such measurements can only work for small ice catchments, as in large ice sheets the grains size gets smaller with longer subglacial fluvial transport distances & would over-estimate the Abrasion component……).
**Added text in *italics/underline*:**

"Despite considerable challenges in observing erosional processes occurring under ice, our understanding of subglacial erosion rates continues to expand. Total glacial erosion rates (i.e., abrasion + quarrying) have been inferred using a variety of *both direct and indirect* approaches (e.g., Hallet et al., 1996; Herman et al., 2021; Koppes, 2022) and are found to generally fall between 0.01 and ≥1 mm yr$^{-1}$; however, higher rates have been measured on short (annual to decadal) timescales (e.g., Koppes and Montgomery, 2009; Cowton et al., 2012). Attempts at separating the components of quarrying and abrasion have been made based on sediment flux measurements (e.g., Loso et al., 2004; Riihimaki et al., 2005), cosmogenic-nuclide inversions across subglacial bedforms (e.g., Briner and Swanson, 1998) and theoretical considerations related to sparsely vs. intensely fractured bedrock (e.g., Anderson, 2014). To date, measurements that isolate the eroded rock volume that can be attributed to quarrying are rare."

98. Mention the distances from the quarried step/face. ".. two from the quarried floor, 1 and 2 m from the quarried step/edge, and one from a polished surface more distal (4m) from the quarried step/edge..".
**Added.**

208. 'four nearby results'. Do these occur on Figure 6? If so, their position should be plotted on that figure. Or on Figure 1. Also, we get no idea of the variability of the total of five abrasion depth values: this should be added – best as a separate, small table.
**These are plotted on Figure 1. Their depths are shown on Figure 1.**

210-213. If reporting only averaged abrasion depth and not volume (see General Comment 2), these lines can go, but a calculation of the average abrasion depth should be written instead. That would be more informative, and less awkward.
**We prefer to derive volumes removed (and not area-averaged depths) from each erosion process as this gets away from implying that quarrying is uniform. The area across which we apply an abrasion depth needs to consider cavities, which we attempted as described in these lines.**

238. Lee face orientation is a poor constraint on ice-flow direction, as it is commonly heavily predicated or influenced by the orientation of the pre-existing fractures (joints), and can thus deviate by up to 40° from the ice-flow direction; see study of Hooyer, et al. (2012). Control of glacial quarrying by bedrock joints. Geomorphology, 153, 91-101. I strongly suggest to take this out.
**Done.**

273. "… processes related to the formation of crescentic gouges (Gilbert 1905)…".
1. Should be Gilbert 1906??
**1906 – thank you!**

2. Although Gilbert (1906) is a great paper, well ahead of its time, there are quite a few more modern relevant references: Harris, 1943; Embleton & King, 1975; Ficker et al., 1980; Wintges,

1985; Prest 1983; Drewry, 1986; Glasser and Bennett, 2004, and Krabbendam et al. 2009. The latter gives a reasonably good overview of the terminology, too (although dealing with a special case)
**Added.**

3. The phrasing here is very vague. I think it would be good and informative to mention the actual process, namely exertion high clast-bed contact forces, exerted by large boulders embedded in basal ice pressing onto the bed. (see also Comment on Line 339).
**Included.**

297-298. It is still unclear to me what the justification for 1790 CE is for the start date of the historic glaciation – and not 1750 or 1700… A short sentence, summarising the previous work would be helpful here.
**See comment below in response to Reviewer 2.**

262. hard crystalline rock, fracture spacing .. AAAH! This should be described in the Setting!
**Moved to higher in the paper where we introduce the study site.**

339. This is the first mention of clast-bed forces, which reads very odd to people not familiar with crescentic gouges. This is why this should be explained near Line 273.
**See above.**

Minor text issues (mainly to clarify things).

Underline means suggested addition to text.
Title: why not mention 'rates'. "In Situ 10Be modelling and terrain analysis constrain quarrying and abrasion rates at Jakboshavn Isbrae, Greenland". Just to set it apart from the more general geomorphological papers around….
**Added 'rates' per suggestion.**

Lines
36. "… rock fragments are entrained in basal ice and pressed..". Clarity..
**To a world expert on ice-bed interactions, pressed is a bit simplistic, but to the general readership, we think it suffices at this introductory section of the paper. Farther below we elaborate based on reviewer suggestion above.**

51. ".. a well constrained, short-duration (c. 200 yr) advance-retreat cycle…"
**It is rare to have an advance/retreat cycle of a glacier so well constrained. Nowhere else in Greenland is it this tight. We are satisfied with this description, but do add more information behind it in another section (see below comment).**

78. possibly mention surface roughness? Mm-scale roughness, or similar. " noted rock surface texture and roughness to distinguish abraded from quarried surfaces…" or similar.
**Adopted.**

83. ".. divot…" this seems to be mainly a rather obscure term originating from golf….? Would "hollow" or similar not be better..? I feel this either needs changing throughout, or explained at first use. Throughout the manuscript…

**Yes indeed a bit golfy, but it's common parlance, not too obscure. How have others described the "scar" left from quarrying? We have not heard "hollow"; we think "divot" gets the point across better.**

84. ".. a sharp transition from rough fractured to smooth abraded surface texture.."
**Changed to surface roughness.**

86. ".. to estimate the chape of the quarried volume (single… "
**Huh?**

89-90. This is awkward phrased: try to improve.
**Reworded: "At another site (Location B; Fig. 3B), there are two adjacent lee steps, each exhibiting a different surface roughness (one rougher, one smoother). Here, we measured the $^{10}$Be concentration at the base of each step to test our hypothesis that different surface roughness relates to quarrying during the historical overriding versus the prior glaciation."**

93-95. Bit of repetition here. Most of Line 93-94 are repeated in next sentence. Why not: "At location A, we sampled and measured 10Be concentrations in five samples…."
**Copy, we condensed to one sentence.**

111. "….(1: freshly fractured exposed surfaces…."
**Done.**

112: "…. that the fresh-appearing fracture surfaces …"
**Done.**

114. first use of gouges can be deleted.
**Check.**

200. "..erosion took place recently."
**Check.**

222-224. Great! I like this!
**Nice!**

223. "… prior main glaciation" (or some other word).
Changed to "during LGM glaciation"

225. ".. a mixture of rough fractured and smooth abraded lee surfaces."
**Check.**

247. "… for establishing the relative contribution.." (Suggestion only)
**Sure that's fine.**

248. ".. on computational models or proxy inferences made from ….". For emphasis…
266-269. I like this!
**Done.**

300. "… but the ratio of abrasion to quarrying and the total depth of glacial erosion over the last historic glaciation would be unaffected."
**Done.**

Good luck with this – I truly hope this gets published. You guys actually scooped us with these methods - hohum, but such is life.
**Thank you! I'd love the chance to learn more about this work you speak of!**

Maarten Krabbendam, Edinburgh, 25 May 2023

**Reviewer 2**
General comments
This is a very good manuscript dealing with glacial erosion at a site on western Greenland. The manuscript is well written and well presented and the research presented should be of interest to a fairly large number of people. In short, this is a great manuscript presenting highly interesting data and using novel techniques to analyze the data on a topic where there is still much to learn. I only have a few points with suggestions for improvements listed below.

Specific comments
1. An underlying assumption for the cosmogenic nuclide modeling that is never properly spelled out in the manuscript is that the bedrock surface started with zero cosmogenic nuclides at the deglaciation 7400 years ago. This assumption should be better spelled out and ideally also motivated somehow. Perhaps it is possible to give some rough estimate of how common or uncommon it is with bedrock samples in the region (or in Greenland in general) having cosmogenic nuclide concentrations that clearly indicate inheritance from cosmogenic exposure prior to the LGM?
**Good point. We added "The premise of this approach requires no $^{10}$Be in these surfaces inherited from prior to the previous glaciation, the LGM in this case. After extensive $^{10}$Be dating in the region of heavily scoured surfaces, inheritance seems absent (e.g., Young et al., 2013)."**

2. The timing of initial burial under ice for the last ice cover period at 1790 CE should be better motivated or explained. It presently seems to come from some references but it would be good to give a brief motivation of how you ended up at the year 1790. In row 68 you even present the duration of ice cover as 220±5 years and five years is a really narrow uncertainty implying that the year 1790 CE should be really well constrained.

**We now go into more detail in the setting by writing: "We infer that ice flowed over our study site for a duration of 220±5 yr. The advance phase timing stems from prior research at an ice-dammed lake (which drained in 1990) that was first dammed (based on varve counts) around 1800 CE (Briner et al., 2011). As in Young et al. (2016), we estimate that the ice had advanced across our study area about a decade prior to it reaching the site of the ice-dammed lake, resulting in our estimate of 1790 CE as the timing of ice arrival at our study area. The retreat of ice from our study site in 2010 CE is based on historical imagery (Balter-Kennedy et al., 2021)."**

**Furthermore, we re-visit the impact of this duration on our conclusions in the discussion section.**

3. In the cosmogenic nuclide modeling you use a linear function for 10Be production from muons and given the shallow depth this seems perfectly fine. However, you should present what muogenic production you use and what it is based on.
**Copy, added.**

4. In Table 2 you present 10Be concentrations and apparent ages and under the table you present necessary data including coordinates, elevation, sample density, 10Be standard, and topographic shielding. However, you do not present the sample thickness and you should include this data so that others can do recalculations without having to guess the sample thickness or ask you about it.
**Sample thicknesses added to table.**

5. In several of the figures, the letter sizing seems a bit odd with large ABC… for the various panels and much smaller letters for some other texts. I believe the figures could look a bit better with a bit more consistent text size and not so large ABC. In Figure 1, I find the acronyms J.I.nb, J.I.sb and J.If a bit strange and unnecessary and I would suggest writing out the full names "Jakobshavn Isbræ" and "Jakobshavn Isfjord" in smaller text and perhaps adding N and S instead of nb and sb. Figure 4 (which is a great figure in many ways!) is a bit too blue for my taste and I would suggest trying to make the ice gray instead of blue. All of these figure comments are however not really important so just disregard them if you disagree.
**We re-drafted Figure 1 to make the text sizing a bit less bizarre! Yes Figure 4 is a bit blue, but we decide to leave it.**

Technical corrections
Row 67: "Balter-Kennedy et al., 201" should be "Balter-Kennedy et al., 2021".
**Done.**

Rows 183-184: I might be wrong, but is it not one "to" too much here: "…to which all further raster analysis was standardized to"?
**Done.**

---

## Editor Decision (ED1)

[revised manuscript text omitted]

**A**

NW | N 40 | NE
W | | E
SW | | SE
S

Location B

Location A

m

■ quarried zones
▦ sediment cover
〰 study area outline
〰 ledges
— fracture

**B**

N

1.4
m removed
0.0

---

## Author Response (AR2)

Response to editor comments

Thanks to Arjen for very helpful comments for fine-tuning a few neglected odds and ends. We adopted all suggestions with the following exceptions:

- We did not add lat/long/north arrow/scale to Figure 1A (map space showing the entire Greenland island outline)
- In the last figure, figure 8, we did not say where specifically the photos were from in terms of adding a symbol on a prior map figure; instead, we wrote that the photos are from the field area, as we did with Figure 2.
- There is a panel C in Figure 5.

Thanks again,
All the best,
Jason Briner